# Worry and Depressive Symptoms in Adolescents with Neurodevelopmental Disorders

**DOI:** 10.3390/ijerph22020185

**Published:** 2025-01-28

**Authors:** Suzanne Stewart, Stephen John Houghton, Ken Glasgow, Leslie Macqueen

**Affiliations:** The Graduate School of Education, The University of Western Australia, 35 Stirling Highway, Perth 6009, Australia; 22083348@student.uwa.edu.au (S.S.); ken.glasgow@uwa.edu.au (K.G.); leslie.macqueen@uwa.edu.au (L.M.)

**Keywords:** adolescents, worry, neurodevelopmental disorders, depressive symptoms

## Abstract

Although worry is a normal cognitive process experienced by adolescents, for a significant number, it can reach intense and uncontrollable levels. If left untreated, these worries can lead to significant mental health problems that are maintained into adulthood. Adolescents with neurodevelopmental disorders (NDDs) may be more prone to cognitive biases (that precede worry) and therefore highly vulnerable to worry. Limited research has examined worry in adolescents with NDDs; however, most studies have focused on measuring anxiety. The present research administered an instrument specifically developed to measure worry to 404 10 to 16-year-old mainstream school-aged adolescents, 204 (123 males, 81 females) of whom had a formally diagnosed NDD. A measure of depressive symptoms was also administered. Confirmatory factor analysis revealed a satisfactory fitting model for worry. Multivariate analysis of variance revealed no interaction effects or main effect for worry according to NDD/non-NDD status. There were, however, main effects for sex, with females scoring significantly higher than males on worry about academic success and the future; worry about peer relationships; combined worry score; and depressive symptoms. The findings of this study offer psychologists and educators a brief validated measure of worry that is suited to mainstream school adolescents with or without NDDs. The wider implications of the findings in the context of education and intervention for students with NDDs are discussed.

## 1. Introduction

Worry is a normal cognitive process during adolescence, with up to 70% of adolescents reporting multiple daily worries that often increase and decrease in number and intensity with the daily environmental pressures from school, peers, and family [1,2,3]. Consisting of relatively uncontrollable thoughts and images regarding potential negative outcomes, worry is repetitive and future-oriented, the content of which often includes how future negative events and experiences might be prevented or managed [4,5,6,7,8]. While moderate levels of worry can serve as an adaptive process [9], for 25% of adolescents, worry can reach intense and uncontrollable levels [1,10]. According to Hirsch and Matthews [11], such pathological levels of worry are maintained by intolerance of uncertainty (i.e., a tendency to react negatively to uncertain situations [12]), emotion dysregulation, and maladaptive beliefs, which if not treated can result in significant mental health problems, absence from school, discontinuing social activities, isolation, and academic dysfunction [13,14,15,16].

Typically, adolescents worry about a range of everyday concerns relating to school, interpersonal and social problems, relationships, health, achievement, social acceptance, psychological wellbeing, exams, appearance, fitting in, the future, climate change, upcoming stressful events, and money problems [1,17,18,19,20,21,22]. During the COVID-19 (SARS-CoV-2) pandemic, some researchers coined the term “COVID stress syndrome”, which consisted of a range of psychological symptoms originating from worries relating to the coronavirus pandemic [23,24]. While this is not surprising, since adolescents were suddenly confronted with threats of infection, illness, the death of loved ones, and thwarted future academic and career prospects [25], worries pertaining to feeling lonely, lacking academic support, academic achievement [26], motivation and completing schoolwork [27], and families and peers/friends [28] were still at the forefront during pandemic times.

Adolescents with neurodevelopmental disorders (NDDs: Specific Learning Disorders [SLDs], Intellectual Disability [ID], Communication Disorders, Autism Spectrum Disorder [ASD], Attention-Deficit/Hyperactivity Disorder [ADHD], and Neurodevelopmental Motor Disorders, including Tic Disorders [29,30]) may be more susceptible to cognitive biases (that lead to worry) and as such more vulnerable to worry [5,6,31] and mental health problems [32]. This susceptibility to the cognitive biases preceding worry is further heightened by the difficulties that adolescents with NDDs experience in reading and responding appropriately to rapid and nuanced social cues as they navigate everyday environments and events [33,34,35,36]. In addition, they tend to interpret everyday ambiguous interpersonal stimuli such as others’ behaviors as threatening or negative [37,38]. There is little research examining worry in adolescents with NDDs, however, with most studies focusing on measuring anxiety. While there is some overlap between the two constructs, each has distinct features. Worry comprises repetitive, relatively uncontrollable thoughts about potential negative life events that begin when an individual experiences an issue whose potential outcome is uncertain but contains the possibility of one or more negative outcomes, the consequences of this being negative emotions such as anxiety. One recent study [39], “the focus of which was on the impact of anxiety” (p. 2468), attempted to show this distinction between worry and anxiety by asking 45 7 to 17-year-old school students on the autism spectrum to respond to 21 statements all preceded by “When I’m worried” (e.g., “When I’m worried it’s harder to do my best schoolwork”; “When I’m worried, I talk less to my friends or other students”). The students sorted the statements according to their own experiences and a Q-sort activity revealed anxiety about academic activities was most highly ranked, followed by friendships.

ADHD, ASD, and SLD are the most commonly presenting NDDs in child and adolescent mental health services (CAMHS) and mainstream schools worldwide [40,41], and the comorbidity within and between these NDDs [42] intensifies the prospect of developing mental health problems such as depression [43,44]. Depression is the leading cause of disease burden in young people [45], and rates of depression are five times greater in those with ADHD [46,47] and highly prevalent in those with ASD (up to 39%) [48,49] and SLD [50]. Worry is a common contributing mechanism to depression and other psychiatric conditions [51,52,53], with research showing it is positively associated with child and adolescent measures of depression [54,55]. Given the degree of psychological distress that individuals with ASD, ADHD, and/or SLD experience [56], who also have more than twice the suicide mortality rate compared to the non-NDD population [57], understanding the impact of the processes that can contribute to and prolong psychological distress is critical.

Understanding the acute pain of negative thoughts in daily life [58] is one of the processes which to date has received scant attention. Worry, as a cognitive function, forms a core foundation for psychological wellbeing and illbeing (internalizing and externalizing disorders) [59] and therefore, increasing knowledge about worry and its impact on students, especially those who are more vulnerable to adverse mental health, is an important step in building supportive environments. School psychologists and educators play a critical role in this, especially through conducting psychoeducational assessments, monitoring student progress [60,61], and providing direct and indirect therapy and associated services to adolescents who face a variety of psychological problems [62]. The lack of available measures and the limited reliability of those that are available [63] have made research and psychoeducational assessment in schools challenging. Worry questionnaires designed for children and adolescents have been drawn from adult conceptualizations of worry and anxiety, and adolescents may not fully comprehend adult-related items [17]. In addition, the factorial structure (i.e., one-factor, two-factor, and viable single higher-order factor structures) and item similarity and redundancy (e.g., “I worry all the time” and “I am always worrying about something”) of even the most popular (and considered gold standard) measure of worry (the Penn State Worry Questionnaire) (PSWQ, [64]) have been identified as contentious [65].

Hunter and colleagues developed the Perth Adolescent Worry Scale (PAWS, [17]), a 12-item self-report measure co-designed with adolescents, including those diagnosed with NDDs. In its initial empirical validation in 2019, 92 adolescents diagnosed with an NDD (ADHD, ASD, SLD) were recruited as part of a sample (N = 835). Two factors, each of 6 items, were identified from an initial 27 items: Factor 1—Worry about Peer Relationships (e.g., others talking about you online, fitting in with other students at school, being judged by friends) and Factor 2—Worry about Academic Success and the Future (e.g., exams or tests, keeping up with schoolwork, what you will be doing when you finish high school, letting your parents down). Demonstrating strong measurement invariance across age and socio-economic status and weak invariance for sex, the PAWS revealed that adolescents with NDDs had significantly lower scores than their neurotypical peers on both Worry about Peer Relationships and Worry about Academic Success and the Future.

Given the paucity of empirical evidence pertaining to worry among adolescents with NDDs, the present study sought to further establish the psychometrics of the PAWS with adolescents with NDDs and test for differences in worries between adolescents with or without NDDs. In doing so, school psychologists and educators will have a short measure with sound psychometric properties for assessing adolescent worry. Consequently, schools will have another measure for evaluating strategies put into place to prevent the onset of adverse mental health.

In line with the aims of the research, three main research questions were formulated:Research Question 1: Does the Perth Adolescent Worry Scale (PAWS) demonstrate satisfactory model fit for adolescents with or without NDDs?Research Question 2: Are there differences in scores for worry about peer relationships, worry about academic success and the future, and depressive symptoms in relation to adolescents with or without NDDs?Research Question 3: Are there differences in scores for worry about peer relationships, worry about academic success and the future, and depressive symptoms according to sex and school grade?

## 2. Materials and Methods

### 2.1. Participants and Settings

The total sample comprised N = 404 adolescents. Of these, 204 (123 males, 81 females) had a diagnosed NDD (ADHD = 76, ASD = 32, SLD = 96) and were Level 1 functioning in mainstream schools. Level 1 indicates that adolescents with an NDD require minimum support (e.g., help with organization or planning) for their everyday independent functioning in school or with their relationships compared to others of the same age and background. Conversely, those with Level 2 or 3 require substantial external support to function independently (e.g., speech therapy, social skills training, full-time aides). These were closely matched on age and gender with 200 neurotypical adolescents (120 males, 80 females) without NDDs. To be included in the NDD sample, students required a formal pediatric diagnosis according to DSM-IV-TR or DSM-5 criteria and had to be following regular mainstream school classes with Level 1 minimal support required to function in daily activities. Table 1 shows the distribution of participants with or without NDDs by school grade level.

The sample was recruited from school grades 6 (ages 10 to 11 years) to 10 (ages 15 to 16 years) in six randomly selected government and non-government Western Australian secondary schools (two in each of three geographical regions) within a 50 km radius of the Perth city center (i.e., the greater Perth area). The schools were in a range of socio-economic status areas (as indicated by their Index of Community Socio-Educational Advantage [ICSEA]). ICSEA is set at an average of 1000 (SD = 100) and the greater the ICSEA score, the higher the level of educational advantage of students who go to the school and vice versa [66]. In this sample, the schools’ ICSEAs ranged from 904 to 1191, thus showing a representative range across schools.

### 2.2. Instrumentation

#### 2.2.1. Adolescent Worry

The Perth Adolescent Worry Scale (PAWS, [17]) is like many other instruments, having been initially generated from a review of the relevant literature and existing instruments. However, unlike other worry instruments, the PAWS was co-designed via a series of focus groups with 10 to 16-year-old adolescents with or without NDDs (n = 36), parents/guardians (n = 16), educators (n = 16), and individual interviews with school psychologists (n = 8). This rigorous procedure identified candidate items that more appropriately reflected concepts for the assessment of worry in adolescents with NDDs. Nevertheless, because of its limited use to date, it is important to further establish the psychometrics of the PAWS.

Initially, the PAWS consisted of 27 items, to which participants self-reported how often (i.e., frequency) they were worried about it (0 = “Never”, 1 = “Sometimes”, 2 = “Often”, 3 = “Always”) and how much worry (i.e., degree of worry) it caused them (0 = “Not at all”, 1 = “A little bit”, 2 = “Somewhat”, 3 = “A lot”). The two scores for each item were also combined and multiplied to produce a single item score. For example, for the item “Exams or tests”, the responses assessing frequency of worry and degree of worry were combined by creating the product of these two estimates (i.e., creating a 0–9 scale for each item). Using this approach, a split-sample exploratory (EFA) and confirmatory (CFA) factor analysis on data from 835 10 to 16-year-olds (317 males, 508 females, 10 unspecified) (including 92 with NDDs) resulted in a 12-item measure representing two factors: Worry about Peer Relationships (6 items) and Worry about Academic Success and the Future (6 items), with acceptable fit to the data (χ^2^ (*df* = 53) = 222.03, *p*< 0.001; CFI = 0.928; RMSEA = 0.088 (90% CI = 0.076, 0.100)) and satisfactory internal reliability for both subscales (α Peer = 0.83; α Academic = 0.88). There was also strong measurement invariance when comparing adolescents with an NDD to those without. In this present study, there was satisfactory internal reliability (Worry about Academic Success and the Future α = 0.89 and Worry about Peer Relationships α = 0.84).

#### 2.2.2. Depressive Symptoms

The Children’s Depression Inventory 2 (CDI 2: SR[S]) [67] is a short and highly valid self-report assessment of depressive symptoms in children and adolescents aged 7–17 years [67]. The CDI 2: SR[S] comprises 12 items, each of which has three separate sentence response options that describe participants’ feelings and ideas over the past two weeks (e.g., “I am sad once in a while”, “I am sad many times”, “I am sad all the time”; CDI 2: SR[S]). The CDI:SR[S]2 reports satisfactory reliability (α = 0.77–0.85) with different age and gender groups [67] and in samples of neurologically diverse Australian youth (α = 0.86 [68]; 0.85 [17]). In the present study, the estimate of reliability was sufficiently high to provide confidence in the use of the CDI 2: SR[S] score (Cronbach’s alpha α = 0.84).

### 2.3. Procedure

Prior to conducting this research, permission was obtained from the Human Research Ethics Committee (2019/RA/4/20/6130) of the administering institution and the State Department of Education for Western Australian (WA) (#D18/0383437). For the one participating non-government school, approval was obtained from the school principal. In addition, approval was obtained from the publishers of the CDI 2: SR[S] to administer it online. Eight randomly selected schools were contacted to gauge their interest in being involved in the research. Schools that agreed to be part of the study identified a senior staff member who provided administrative support, including liaising with the researcher to ensure a consistent and standardized approach to the administration of the online Qualtrics survey. Letters of introduction, information sheets, and consent forms were distributed to parents/guardians of children in each school (school grades 6–10) and approximately 70% of parents agreed for their child to be involved.

Maintaining the anonymity of the participants and the confidentiality of their responses was important, so prior to beginning the survey in 2021, all students were briefed by their liaising educator on the aims of the research and assured that their responses were confidential. They were then informed that they could stop and withdraw whenever they wished. Each student was then given a unique identification code so they could log into the Qualtrics survey. The students were told that this unique code made sure that all information they provided was confidential and that only the researcher would see their answers to the questions. The school principal maintained a list of these codes against participating students’ names.

Although NDD status was determined via official diagnoses, the first section of the survey asked students to self-report if they had been diagnosed with ADHD, ASD, and/or SLD. These self-reported data were then checked for accuracy by the school principal and/or school psychologist against official school records. A check was also made at this time for students with a formal diagnosis who may have failed to report this. The surveys were administered to students in groups of 10 to 25 during regular school hours at times determined by each school. On average, the survey took 20–25 min to complete. Accommodations were made for any students requiring additional support during the administration.

### 2.4. Data Analysis

Given the PAWS has previously been utilized with a relatively small sample of adolescents with NDDs [17], a confirmatory factor analysis using AMOS 27 [69] was conducted to verify the fit of the original 12-item PAWS scales. Four indices were utilized to assess the “goodness of fit” of the model: the Comparative Fit Index (CFI) and Non-normed Fit Index (Tucker–Lewis Index: TLI) (CFI and TLI: above 0.95 indicates good fit, above 0.90 indicates adequate fit), the Root Mean Square Error of Approximation (RMSEA: 0.05 or less indicates good fit, 0.08 or less indicates adequate fit), and Chi-square (non-significant values represent good fit). MacCullum and Austin [70] strongly recommended the routine use of the RMSEA because (i) it is adequately sensitive to model misspecification, (ii) commonly used interpretive guidelines would appear to yield appropriate conclusions regarding model quality, and (iii) it is possible to build confidence intervals around RMSEA values. To test for differences, a Multivariate Analysis of Variance (MANOVA) was performed on the dependent variables of the PAWS and the CDI 2: SR[S] according to NDD status (NDDs, Non-NDDs), sex (Male and Female), and school grade (Grades 6 to 10).

## 3. Results

The two-factor 12-item measurement model for the PAWS displayed a satisfactory fitting model: χ^2^ (*df* = 53) = 198.06, *p* = < 0.001, CMIN = 3.73, CFI = 0.92, TLI = 0.90, RMSEA = 0.065 (90% CI: 0.058, 0.078). The correlation between the two factors was 0.66 and the standardized regression weights revealed that the factor loadings ranged from 0.45 (keeping up with schoolwork) to 0.81 (being judged by friends). Model fit was adequate for adolescents with NDDs (CMIN = 3.14, CFI = 0.90, TLI = 0.88, RMSEA = 0.07 (90% CI: 0.062, 0.087)) or without NDDs (CMIN = 2.61, CFI = 0.93, TLI = 0.89, RMSEA = 0.08 (90% CI: 0.071, 0.107)). This indicates that the PAWS is appropriate for conducting research that examines the worries of adolescents with or without NDDs. The correlations between Depressive Symptoms (CDI 2: SR[S]) and Worry about Peer Relationships (0.48), Worry about Academic Success and the Future (0.38), and Total Worries (0.48) were all positive in the expected direction and significant (*p* < 0.01). That is, the experience of worry is associated with depressive symptoms and individuals who worry tend to have higher scores for depressive symptoms.

A Multivariate Analysis of Variance (MANOVA) was performed on the dependent variables of the PAWS and the CDI 2: SR[S] according to NDD status (NDDs, Non-NDDs), sex (Male and Female), and school grade (Grades 6 to 10). Box’s Test of Equality was significant (*p* < 0.05); however, this test is very sensitive to data files that are large and can detect even small departures from homogeneity. Moreover, it can be sensitive to departures from the assumption of normality. In this present study, the *F* value was small (2.07). To address this, a Bonferroni adjusted alpha level of <0.025 was used for the PAWS.

There was no significant multivariate interaction effect for NDD Status x sex x school grade: Wilks’ Lambda, *F* (16, 1033.245) = 0.723, *p* = 0.772, partial η^2^ = 0.008. There were significant multivariate main effects, however, for sex (Wilks’ Lambda, *F* (4, 338.00) = 5.310, *p* < 0.001, partial η^2^ = 0.059) and school grade (Wilks’ Lambda, *F* (16, 1033.245) = 1.950, *p* < 0.05, partial η^2^ = 0.022). There was no main effect for NDDs status (Wilks’ Lambda, *F* (4, 338.00) = 2.669, *p* = 0.032, partial η^2^ = 0.031). The Univariate *F* and observed means for the main effect of sex are presented in Table 2 (no main effects were evident for school grade level when the Univariate effects were examined.) Using the Bonferroni adjusted alpha level of <0.025, the PAWS (Worry about Academic Success and the Future; Worry about Peer Relationships; and the Total Combined Worry score) and depressive symptoms were all significant for sex. NDD Status and school grade level did not reach levels of significance. As can be seen in Table 2, depressive symptoms were significant (*p* < 0.01), with females scoring higher than males on the CDI 2: SR[S] and the other dependent variables. Thus, while the scores for worry and depressive symptoms did not differ for adolescents with or without NDDs, they did according to sex.

## 4. Discussion

The mental wellbeing of adolescents is a significant public health concern, which is unsurprising given the high rates of problematic mental health globally [71]. According to the World Health Organization [72], approximately 166 million 10–19-year-olds worldwide have a diagnosed mental disorder [73] and 1 in 7 suffer unrecognized or untreated mental health challenges [72]. Adolescents with NDDs are at increased risk of developing adverse mental health compared to their neurotypical peers [43,44,74,75,76,77], and a key foundation in the development and maintenance of these mental health problems is worry [59]. As worry predicts poor mental health [28], developing a greater understanding of it offers potential for intervention, especially among vulnerable populations such as individuals with NDDs. However, research examining worry among adolescents has been limited to date, especially among neurodiverse populations.

As in previous research, and just like their neurotypical peers, adolescents with NDDs in the present study worried about everyday things concerning school, interpersonal and social problems, relationships, achievement, exams, appearance, and fitting in [17,19,22,78]. Moreover, these worries were like those of adolescents from other countries (e.g., [27,28]). The present finding of no differences in levels of worry among Western Australian adolescents with or without NDDs is like that found in earlier work (e.g., [39]) but is different to that found by [17] employing the same measure as used here. One possible interpretation for this absence of difference may be the larger sample of adolescents with NDDs generated in the present study compared to that in the earlier study [17].

Adolescents with NDDs are vulnerable to peer-related isolation and emotional dysfunction [79] because of the major problems they experience in constructing social relationships with their peers, especially reciprocated friendships, in daily social and academic interactions [48,80,81,82,83,84,85]. This is further compounded because adolescents with ASD, ADHD, and SLD have negative perceptions about themselves, their friendships, and peer networks compared with their neurotypical peers [86]. Even when in friendships, those with NDDs may be vulnerable to being treated unfairly because they find it hard to understand the nuances of emotion presentation [87]. This makes them particularly vulnerable to developing negative thought patterns and mental health problems such as depression [43]. However, no differences in levels of worry and/or depressive symptoms were found between adolescents with or without NDDs in the present study The absence of research examining worry in adolescents with NDDs makes comparisons with the present findings difficult; however, the findings regarding depressive symptoms run contrary to previous research that identified higher prevalence rates among those with NDDs [46,47,48,49,50].

Intolerance to uncertainty is a common observation in individuals with NDDs, especially adults (see [56]), and intolerance of uncertainty-based models of worry postulate that individuals high in this will be more inclined to engage in worry (see [88]). However, unlike adults with NDDs, who tend to encounter uncertainty in daily environmental demands in different situations, schools provide adolescents with NDDs with routine or ‘sameness’ and this may contribute to allaying the development of worries, especially about uncontrollable or unpredictable events. This may in some way explain the finding of no differences in worry between adolescents with or without NDDs in the present study.

Significant associations were found in the same positive direction between Worries about Peer Relationships and Worries about Academic Success and the Future and depression in the present research. This aligns with other research with adolescents [54,55] and adults that points to a correlation between worry and depression, even after controlling for anxiety [89,90]. However, adolescents with or without NDDs did not differ on levels of depression in the present study, which may be related to the earlier point made about the larger sample size of adolescents with NDDs in this study.

Rather, where differences were evident in this present study was between males and females, with females reporting greater levels of worry about academic success and the future, worry about peer relationships, total worries, and depressive symptoms. During adolescence, worry is a key contributor to the emergence and continuation of mental health problems, and the present findings support previous work showing that females report significantly higher levels and greater frequencies of worry [78,91] and significantly more internalizing disorders (e.g., depression and anxiety) [92] than males. That adolescent females, when compared to their male counterparts, are twice as likely to experience depressive episodes [93] suggests that further research untangling the complex dynamics of worry and depression is warranted [92]. This should also examine gender differences in emotion expression (i.e., what adolescents show externally, such as facial, vocal, and postural expressions to communicate their internal emotional states to others). Research shows that early gender-related emotion expression patterns can predispose children and adolescents to the development of psychopathology, with girls who present exaggerated female patterns of coping with stressors via high sadness and anxiety being at risk for internalizing distress and developing depression and anxiety (for a review see [94]). In addition, research should also investigate the potential role of social camouflaging because it is related to females and negative mental health outcomes (see [95] for a review).

Previous research by Jefferson et al. [96] and Miller-Lewis [97] identifies schools can be ideal places for interventions that alter adolescents’ maladaptive interpretations (worries) and lead to a lower frequency of mental health problems. Qualter [98] added that schools may provide valuable information for cost-effective, efficacious, and tailor-made early interventions that can be delivered en masse. This is an important consideration in the light of the present findings because worry can become extreme if left untreated over time. This can lead to considerable mental health problems, school absenteeism, discontinuing social activities, and poor academic functioning [13,16], which can result in an adult population burdened by social and economic challenges in the long term [72]. If schools are to be involved, however, they must utilize instruments that are reliable and appropriate for the contexts in which they are administered to inform early intervention [99]. Adopting such an approach will inform an evidence-based approach critical to guide school-based prevention and intervention efforts [100,101,102]. The PAWS provides school psychologists with such an instrument for screening cohorts of adolescents to detect excessive levels of worry for potential intervention. As in previous research [17], the PAWS provided strong evidence about its psychometric properties in this present research study. As such, it seems ideal for use in schools to provide quality data to identify trends, determine the effectiveness of interventions, and reveal emerging challenges [60,103], all of which inform school policies.

## 5. Conclusions

Although the present research was conducted with a systematic and rigorous approach, there are limitations which must be acknowledged. The research focused on Western Australian adolescents diagnosed with ASD, ADHD, and/or SLD. Hence, the findings cannot be generalized to the wider population of other NDDs, such as intellectual impairment, motor disorders, or communication disorders. In addition, it is highly recommended that data collection should be from multi-informants [104]. The data in this study were self-reported. However, third parties such as educators or parents have considerable difficulty perceiving the internal subjective dispositions of their children (i.e., worry) and adolescents tend to find it difficult to report their internal states to their parents and educators [105]. Highly reliable self-report measures such as those used here (e.g., the PAWS and CDI 2: SR[S]) may, however, elicit more valid responses from adolescents. In addition, the psychometrics of the relatively new PAWS was tested in the present research.

The present research appears to be the first study to specifically examine worry with a large sample of adolescents diagnosed with NDDs and to compare these to neurotypical peers. As such, the findings add to the limited evidence currently available. This is important because adolescents with NDDs constitute a large proportion of global disability [106], and the growing numbers of adolescents with NDDs in mainstream schools is providing unique challenges to educators (see [107]). Mainstream classroom educators report feeling ill prepared and insecure teaching adolescents with NDDs (especially ASD) [108,109]. Furthermore, many mainstream educators hold negative perceptions of adolescents with NDDs, especially if they present with low levels of academic achievement and emotional and social development [109]. The present finding that adolescents with NDDs worry about the same academic and peer-related things as their neurotypical peers may alleviate the insecurities and attitudes that mainstream educators hold about adolescents with NDDs and in doing so go some way towards reducing barriers to creating inclusive classroom environments.

The present findings also add to the field by suggesting that whole school programs to reduce negative thought processes can be developed and utilized by all adolescents, irrespective of NDD status, thereby reducing the stigma often associated with the delivery of separate programs for different adolescents in mainstream schools.

## Figures and Tables

**Table 1 ijerph-22-00185-t001:** Distribution of participants with or without NDDs by school grade level.

School Grade Level						
Year Group	6	7	8	9	10	Total
NDD	35	11	79	31	48	204
No NDD	24	19	61	43	53	200
Total	59	30	140	74	101	404

**Table 2 ijerph-22-00185-t002:** Univariate *F* statistics, observed means, and standard deviations for worry and depressive symptoms with sex as the independent variable.

	Mean Square	*F*	*p*	Partial η^2^	Males Mean (SD)	Females Mean (SD)	Total Mean (SD)
Depressive Symptoms	178.783	8.997	0.003 *	0.026	5.87 (4.53)	7.21 (4.53)	6.41 (4.51)
Worry—Academic Success/Future	64.139	10.050	0.002 **	0.029	2.67 (2.49)	3.70 (2.63)	3.08 (2.60)
Worry—Peer Relationships	57.001	18.751	0.001 **	0.052	1.19 (1.65)	1.71 (1.62)	1.45 (1.79)
Worry Total Combined Score	35.601	14.804	0.001 **	0.042	1.61 (1.45)	2.31 (1.70)	1.89 (1.59)

* *p* < 0.01; ** *p* < 0.125.

## Data Availability

Individuals may contact the corresponding author, Professor Stephen Houghton (SH), at stephen.houghton@uwa.edu.au, where the data will be made available upon reasonable email request.

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
