# Peer review of "Worry and Depressive Symptoms in Adolescents with Neurodevelopmental Disorders"

_ijerph, 2025, doi:10.3390/ijerph22020185_

Round 1

Reviewer 1 Report

Comments and Suggestions for Authors

Thank you for providing me with the opportunity to review your manuscript.

The topic of this study is significant as it delves into common problems experienced during adolescence. Adolescent mental health is a critical issue that necessitates appropriate and timely attention.

This study highlights the importance of considering symptoms and offering educators, counselors, and psychologists insights into prevention and intervention strategies to address these issues effectively.

The paper is well-written, drawing support from recent research in the field under investigation and employing a methodological design that aligns with the study's objectives. The methodological procedures are well-described, accompanied by a clear depiction of the study's context.

The findings are relevant as they offer empirical evidence regarding worry and depressive symptoms, emphasizing the urgent need for prevention and intervention measures.

Females have reported higher levels of worry regarding academic success, future prospects, peer relationships, total worries, and depressive symptoms. The authors may consider discussing the differences in emotional expressions between males and females.

Author Response

Reviewer 1

We are grateful to Reviewer 1 for the positive comments.

Response 1. In the light of the comment about the differences in emotional expressions between males and females we have added the following additional text on page 8 in the discussion:

This should also examine gender differences in emotion expression (i.e., what adolescents show externally such as facial, vocal, and postural expressions to communicate their internal emotional states to others). Research shows that early gender-related emotion expression patterns can predispose children and adolescents to the development of psychopathology with girls who present exaggerated “female” patterns of coping with stressors via high sadness and anxiety, at risk for internalizing distress and developing depression and anxiety (for a review see [94]). 

Reviewer 2 Report

Comments and Suggestions for Authors

RE: Manuscript IJERPH-3378194: Worry and depressive symptoms in adolescents with neurodevelopmental disorders

COMMENTS FOR THE AUTHOR:

I enjoyed reading the manuscript, which examines worry among adolescents diagnosed with NDDs. This study addresses an important and timely topic area and addresses a gap in research as it relates to a specific population. The methodology and analysis are thoroughly outlined, which is critical to presenting quantitative research. A particular strength of this research is the methodology. My major concern is with the introduction section and outlining the purpose of this study. I’ve listed specific feedback based on page below.

Abstract

Sentence three, remove “mental health problems” as this was previously stated in sentence two.

Sentences four and five, rephrase for clarity. I suggest combining these two sentences together.

Overall, I suggest revising the abstract for spelling (i.e., lower case “factor analysis” and “sex”).

Introduction

Page 1: “… of adolescents, worries can reach intense…”, worries should be singular here.

Page 2: Missing the closing parenthesis after “…Tic Disorders [29.30])…”.

Page 2: Remove underscore between “they_navigate”.

Page 2: The authors make the case that many studies focus on anxiety, rather than worry; however, only cite one article in the introduction. It is unclear on how the study discussed measured anxiety using the word “worry.” To strengthen the importance of this study, the authors should consider adding a few sentences describing the difference between worry and anxiety and clarifying how this study assesses for anxiety using “worry.”

Page 2: Combine the paragraph beginning with “Given the degree…” with the paragraph above for flow and clarity.

Page 2: Combine the paragraph beginning with "School psychologists….” with the paragraph above flow and clarity.

Page 2: Revise the sentence “worry questionnaires in children…” for clarity. Consider replacing “in” with “for” or “Worry questionnaire designed for children…”.

Page 3: The sentence needs to be completed. The factorial structure of assessment tools has been identified as what?

Page 3: Ensure citation is correct. “…(Penn State Worry Questionnaire [PSWQ], 64).

Page 3: Add the word “diagnosed” to the sentence “…including those diagnosed with NDDs).”

Page 3: Rephrase this sentence for clarity, “…92 adolescents with NDDs…”. One possibility is to rephrase the sentence to say “… 92 adolescents diagnosed with an NDD…”

Page 3: Rephrase the sentence to say “… for sex, the PAWS…”.

Before moving into the Methods and Materials section, include a short paragraph about this study (i.e., rationale, purpose, hypothesis (if appropriate), etc.).

Materials and Methods

Page 3: “…had a diagnosed Neurodevelopmental Disorder (NDD)”, since NDD was previously described, defining the acronym is not needed and the authors can use NDD.

Page 3: Describing what a “Level 1” means in this sentence is needed to provide context for the reader.

Page 3: For clarity, I suggest using ‘diagnosed with an NDD’ throughout the manuscript.

Page 4: “…90% confidence interval…”, confidence interval can be abbreviated here.

Page 4: “…neurodevelopmental disorders”, for consistency, NDD can be used here.

Page 5: “… (…TLI) (CFI and TLI…) …”, the two parentheses can be combined here using a semi-colon. For example, (…TLI; CFI and TLI…).

Page 3: For transparency, the authors should consider adding the dates of data collection to this section.

Results

Overall, the results section is clear and well-written. However, one sentence at the end of each paragraph interpreting the results would strengthen this section and provide clarity for the reader.

Discussion

The first paragraph would be better suited in the introduction to strengthen the authors’ argument for why this study.

Page 7: “…autistic adults…”, rephrase to use person-first language. For example, adults diagnosed with autism.

Page 7: “It is possible therefore…”, for clarity and flow, I suggest revising to “It is therefore possible that adolescents with..."

Page 7: “During adolescence…”, missing comma after adolescence.

Page 7: “… (e.g., depression and anxiety…”, missing the closing parenthesis after anxiety.

Page 7: For readability, the authors might want to consider revising the sentence “According to [102, 103] …”. A suggestion is “Previous research identifies schools can be…”.

Page 8: The sentence “The PAWS provides school psychologists and educators…”, the way this sentence currently reads is that educators can also administer the PAWS, which I don’t think the authors are trying to say. Perhaps the authors are conveying educators and school psychologists benefit from the results of the PAWS? Please revise this sentence for clarity.

Conclusions

Page 8: For consistency, use ASD, ADHD, SLD instead of spelling out the words.

Page 8: Remove the words “In conclusion” and begin the sentence with “This present…”.

Page 8: For consistency, replace “young people” and “students” with “adolescents”.

Page 8: For consistency, replace “teachers” with “educators”.

Author Response

Reviewer 2

Query 1. Abstract

Response 1. Sentence three - we have removed “mental health problems” as this was previously stated in sentence two.

Response 2. We have combined sentences four and five and rephrased for clarity.

Response 3. The abstract has been revised overall in terms of spelling (i.e., lower case “factor analysis” and “sex”).

Query 2. Introduction

Response 1. On Page 1 we have rewritten “… of adolescents, worries can reach intense…”, to reflect the singular (i.e., worry)

Response 2. On Page 2 we have closed the parenthesis after “…Tic Disorders [29.30])…”.

Response 3. On Page 2 we have removed the underscore between “they_navigate”.

Response 4. We have added a few sentences describing the difference between worry and anxiety as follows:

While there is some overlap between the two constructs, each has distinct features. Worry comprises repetitive, relatively uncontrollable thoughts about potential negative life events that begin when an individual experiences an issue whose potential outcome is uncertain but contains the possibility of one or more negative outcomes. The consequences of this being negative emotions such as anxiety.

We have also reworded the following paragraph as follows:

One recent study [39], “the focus of which was on the impact of anxiety” (p. 2468), attempted to show this distinction between worry and anxiety by asking 45, 7 - 17 year-old school students on the autism spectrum to respond to 21 statements all preceded by ‘When I’m worried’ (e.g., ‘When I’m worried it’s harder to do my best schoolwork’; ‘When I’m worried, I talk less to my friends or other students’). The students sorted the statements according to their own experiences and a Q-sort activity revealed anxiety about academic activities was most highly ranked followed by friendships.

Response 5. On Page 2 we have combined the paragraph beginning with “Given the degree…” with the paragraph above for flow and clarity.

Response 6. On Page 2 we have combined the paragraph beginning with "School psychologists….” with the paragraph above for flow and clarity.

Response 7. On Page 2 we have revised the sentence “worry questionnaires in children…” as suggested by replacing “in” with “Worry questionnaire designed for children…”.

Response 8. On Page 3: The sentence needs to be completed. The factorial structure of assessment tools has been identified as what?

This sentence has been slightly amended as follows: In addition, the factorial structure (i.e., one factor, two-factor, and viable single higher order factor structures) and item similarity and redundancy (e.g., ‘I worry all the time’ and ‘I am always worrying about something’) of even the most popular (and considered gold standard) measure of worry (the Penn State Worry Questionnaire) [PSWQ, 64] have been identified as contentious [65].

Response 9. We have checked that the citation on Page 3: (Penn State Worry Questionnaire [PSWQ], [64]) is correct.

Response 10. On Page 3 we have added the word “diagnosed” to the sentence “…including those diagnosed with NDDs).”

Response 11. On Page 3 we have rephrased the sentence “…92 adolescents with NDDs…”. to “… 92 adolescents diagnosed with an NDD…”.

Response 12. On Page 3 we have rephrased the sentence as recommended to say “… for sex, the PAWS…”.

Response 13. Reviewer 3 made a similar point about the final paragraph before the method section and so to achieve a balanced outcome for both reviewers points we have rewritten this paragraph as:

Given the paucity of empirical evidence pertaining to worry among adolescents with NDDs the present study sought to further establish the psychometrics of the PAWS with adolescents with NDDs and test for differences in worries between adolescents with or without NDDs. In doing so, school…..

Query 3. Materials and Methods

Response 1. On Page 3 we have used the acronym NDD as suggested, since NDD was previously described.

Response 2. On Page 3: We have provided more information about what “Level 1” means as follows:

Level 1 indicates that adolescents with an NDD require minimum support (e.g., help with organization or planning) for their everyday independent functioning in school or with their relationships compared to others of the same age and background. Conversely, those with Level 2 or 3 require substantial external support to function independently (e.g., speech therapy, social skills training, full-time aides).

Response 3. As suggested, we have used ‘diagnosed with an NDD’ on Page 3 and throughout the manuscript.

Response 4. As suggested, on Page 4 “…90% confidence interval…”, has been abbreviated.

Response 5. On Page 4: “…neurodevelopmental disorders”, has been replaced with NDD for consistency.

Response 6. Page 5: (…TLI) (CFI and TLI…) …”, Our apologies we could not interpret this point being made in the text regrading combining the two parentheses using a semi-colon. We would prefer to keep the reporting in text as it currently is since this is a conventional way of reporting indices.

Response 7. Page 3: We have clarified data collection as follows:

In its initial empirical validation in 2019

And on page 5 … prior to beginning the survey in 2021….

Query 4. Results

Response 1. In line with the suggestion (and we are grateful for the comment that our results section is clear and well-written) we have added one sentence at the end of each paragraph:

This indicates that the PAWS is appropriate for conducting research that examines the worries of adolescents with or without NDDs.

That is, the experience of worry is associated with depressive symptoms and individuals who worry tend to have higher scores for depressive symptoms.

Thus, while the scores for worry and depressive symptoms did not differ for adolescents with or without NDDs, they did according to sex.

Query 5. Discussion

Response 1. We think that the first paragraph is better placed in the discussion (rather than the introduction) because it reiterates the current world situation, especially for those with NDDs and then sets up the discussion about the present study findings. Therefore, we would prefer to leave this as is.

Response 2. On Page 7 “…autistic adults…”, has been rephrased to person-first language (adults diagnosed with autism).

Response 3. As suggested, on Page 7 we have amended “It is possible therefore…” to “It is therefore possible that adolescents with..."

Response 4. On Page 7 we have inserted the missing comma after adolescence - “During adolescence, …”

Response 5. On Page 7 we have closed the parenthesis after anxiety (e.g., depression and anxiety).

Response 6. As suggested, on Page 7 we have revised the sentence “According to [102, 103] …”. to “Previous research identifies schools can be…”.

Response 7. For clarity, on Page 8 the sentence “The PAWS provides school psychologists and educators…”, has been amended to say that The PAWS provides school psychologists……

Query 5. Conclusions

Response 1. As suggested, on Page 8 ASD, ADHD, SLD have been used instead of words.

Response 2. On Page 8 the words “In conclusion” have been removed, and as replaced with “This present…”.

Response 3. As suggested, on Page 8 and throughout the manuscript “young people” and “students” has been replaced with “adolescents” for consistency.

Response 4. As suggested, on Page 8 “teachers” has been replaced with “educators"

Reviewer 3 Report

Comments and Suggestions for Authors

Dear Authors,

I found your article interesting to read.

I have a few comments I would like you to take into consideration.

In the Introduction section I think the last passage should be rewritten because it sounds like a conclusion, rather than the aim of the research.

In the Introduction section please explain the sentence: “Demonstrating strong measurement invariance… adolescents with NDDs scored significantly lower than their neurotypical peers on both factors.” It is not clear whether this means that they had lower worry score or they worried less.

In the Methods section please explain Level 1 functioning since other countries do not use such categories of school functioning

In the Discussion section I find the section describing similar (reference 39) and dissimilar results (reference 17) unclear. This section should be rewritten. Especially since you reported in the Introduction section result for reference 17 that NDDs adolescents scored significantly lower, and in this section, you reported for the same reference 17 the opposite result - significantly higher scores.

In the Discussion section I find the comment on social camouflaging as a possible explanation of non- significant difference in results of worry level between neurotypical and NDDs adolescents contradicting to the conclusion that this measuring instrument is valid and useful for schools. I think you should comment that no differences between neurotypical and NDDs adolescents might be the result of greater sample used in this study than in earlier research. If an adolescent can camouflage the result of measuring instrument, I can not find this instrument useful in this age group.  

In the discussion section the sentence: “According to (102,103) schools…” you should add information according to who – the name of the researcher, not only reference number.

In the References section there are some double reference numbers in References 83, 86, 87, 88, 89.

With kind regards.

Author Response

Reviewer 3

Thank you for the positive comments.

Response 1. As requested, the last passage has been rewritten as an aim. We have also combined this with a similar point raised by another of the reviewers. To achieve a balanced outcome between two reviewers comments we have rewritten this paragraph as:

Given the paucity of empirical evidence pertaining to worry among adolescents with NDDs the present study sought to further establish the psychometrics of the PAWS with adolescents with NDDs and test for differences in worries between adolescents with or without NDDs. In doing so, school…..

Response 2. As requested, in the Introduction section the sentence “Demonstrating strong measurement invariance… adolescents with NDDs scored significantly lower than their neurotypical peers on both factors” has been rewritten as follows:

….NDDs had significantly lower scores than their neurotypical peers on both Worry about Peer Relationships and Worry about Academic Success and the Future.

Response 3. (One other reviewer asked for similar information). In the Methods section we have included more description about level 1 as follows:

Level 1 indicates that adolescents with an NDD require minimum support (e.g., help with organization or planning) for their everyday independent functioning in school or with their relationships compared to others of the same age and background. Conversely, those with Level 2 or 3 require substantial external support to function independently (e.g., speech therapy, social skills training, full-time aides).

Response 4. We are grateful for you pointing out we reported incorrectly and have rewritten this correctly as follows:

The present finding of no differences in levels of worry among Western Australian adolescents with or without NDDs is like that found in earlier work [e.g., 39], but is different to that found by [17] employing the same measure as used here. Unlike research such as that cited earlier where anxiety was the measured outcome, the Hunter et al. [17] measure was specifically developed to assess adolescent worry, and this revealed that adolescents with NDDs scored significantly lower than neurotypical adolescents for Worries about Peer Relationships and Worries about Academic Success and the Future.

Response 5. As suggested, we have deleted the reference (and paragraph) to social camouflaging as a possible explanation of non- significant differences, and have placed the finding (of no differences) in the context of the larger sample size:

One possible interpretation for this absence of difference may be the larger sample generated in the present study compared to that in the earlier study [17].

Response 6. Another reviewer suggested we rewrite “According to (102,103) as “Previous research [102,103] …. To accommodate the previous reviewer and your suggestion for citing the researchers we have amended this as follows:

Previous research by Jefferson et al. [102] and Miller-Lewis [103] identifies schools can be ideal places …

Response 7. The double reference numbers in References 83, 86, 87, 88, 89 have been corrected.

Reviewer 4 Report

Comments and Suggestions for Authors

Thank you for this timely paper. As number of children with NDD increases, we need better understanding of their functioning, both related to disorder and relatively independent.

1. On page 2 paragraph "School psychologists and educators play a critical role in this, especially through conducting psychoeducational assessments, monitoring student progress [60, 61] and providing direct and indirect therapy and associated services to young people who face a variety
of psychological problems [62]." seems unconnected to the rest analysis. It would be great if some logical connections were added.

2. It would be great if you gave some details about Level 1 functioning (Methods and materials section) for readers from the countries that have alternative ways of NDD assessment.

3. Could you somehow comments why amount of males is significantly higher in both groups than girls? Was it on purpose? Occasionally? Maybe its demographic specifics?

4. There is no aim or research questions in the text, so it is quite complicated to estimate the logic and relevance of the procedures. It would be great if you clarified it.

5. I got confused what was the purpose of the study (as mentioned in the previous comment, there was no direct aim in the text, so I had to guess, based on the title and results). So, from the title I could expect that the paper will be analyzing specifics of worries and depressive symptoms in adolescents with NDD, but from results section it feels like the aim was to validate short version of PAWS. You need to clarify that.

6. It is now quite clear why would you describe in such details (section 2.1.1) the development of PAWS as long as it is already published.

7. Than you describe CFI indexes first in section 2.1.1 and next in section 3. Is it really necessary?

8. You could expand your discussion section with some insights about what your results add to the field. As for now it feels like you confirmed that PAWS is a good instrument to measure worries, and it is good for both neurotypical and NDD adolescents. But mainly we could conclude that from the previous paper. The amount of NDD adolecents in previous study was smaller, but not that much smaller to make a new study.

9. It would be nice if describing the sample you would give proportion of participants with specific types of NDD, so it would make the composition of the sample more clear.

Author Response

Reviewer 4

We are grateful for the positive comments.

Response 1. On page 2 we have amended the paragraph beginning “The lack of available measures, and the limited reliability of those that are available [63] ….”  to include “…. psychoeducational assessment in schools as challenging”. We have run this onto the preceding paragraph beginning "School psychologists and educators play a critical role in this, …… to try and increase relevance.

Response 2. about Level 1 functioning. Two other reviewers made similar points and so we have included the following explanation:

Level 1 indicates that adolescents with an NDD require minimum support (e.g., help with organization or planning) for their everyday independent functioning in school or with their relationships compared to others of the same age and background. Conversely, those with Level 2 or 3 require substantial external support to function independently (e.g., speech therapy, social skills training, full-time aides).

Response 3. The sample was generated from the returned consent forms and more males is simply the result of this. We can include a comment on this if you think it is beneficial.

Response 4. As requested, we have included three main research questions that follow the aim of the study:

In line with the aims of the research three main research questions were formulated:

Research Question 1: Does the Perth Adolescent Worry Scale (PAWS) demonstrate satisfactory model fit for adolescents with or without NDDs?

Research Question 2: Are there differences in scores for worry about peer relationships, worry about academic success and the future, and depressive symptoms in relation to adolescents with or without NDDs?

Research Question 3: Are there differences in scores for worry about peer relationships, worry about academic success and the future, and depressive symptoms according to sex and school grade?

Response 5. In response to a similar comment from another reviewer we have rephrased the final paragraph before the method as follows:

The aim of this present study is to further establish the psychometrics of the PAWS with adolescents with NDDs and test for differences in worries between adolescents with or without NDDs.

Response 6. It is not quite clear why would you describe in such details (section 2.1.1) the development of PAWS as long as it is already published.

The reason for this is to inform readers that the PAWS was co-designed with adolescents and appears to be the only worry measure that has been developed in this way. Showing this, particularly in the light of other measures purporting to measure worry that have not followed such a rigorous pathway, is important. If the reviewer believes that removing the information would be beneficial, we are happy to respond positively.

Response 7. Our apologies for the duplication of CFI indexes. We have deleted the CFI indexes in 2.1.1 and left only the internal reliabilities. The CFI indexes are now only reported in 3. Results

Response 8. The final section of the conclusion section has been rewritten to provide insights about what our results add to the field as follows:

This present research appears to be the first study to specifically examine worry with a large sample of adolescents diagnosed with NDDs and to compare these to neurotypical peers. As such the findings add to the limited evidence currently available. This is important because adolescents with NDDs constitute a large proportion of global disability [112], and the growing numbers of adolescents with NDDs in mainstream schools is providing unique challenges to educators [see 113]. Mainstream classroom educators report feeling ill prepared and insecure teaching adolescents with NDDs (especially ASD) [Suhrheinrich, 2011; Gómez-Marí et al., 2021, 2022]. Furthermore, many mainstream educators hold negative perceptions of adolescents with NDDs, especially if they present with low levels of academic achievement and emotional and social development [GómezMarí et al., 2021, 2022]. The present finding that adolescents with NDDs worry about the same academic and peer related things as their neurotypical peers may alleviate the insecurities and attitudes that mainstream educators hold about adolescents with NDDs and in doing so go some way to reducing barriers to creating inclusive classroom environments.

We have also amended the subsequent text as:

The present findings also add to the field by suggesting that whole school programs to reduce negative thought processes can be developed and utilized by all adolescents, irrespective of NDD status, thereby reducing the stigma often associated with the delivery of separate programs for different adolescents in mainstream schools.

Response 9. As requested, we have provided the following specific types of NDD for the sample:

(ADHD = 76, ASD = 32, SLD = 96) and were Level 1 functioning in mainstream schools.

Round 2

Reviewer 3 Report

Comments and Suggestions for Authors

Dear Authors,

I think you managed to improve your article.

I have some suggestions.

In the Results section there is no CDI results presented according to NDD status.

In the Discussion section, passage 2, after mentioning reference 17, the sentence starting with: "Unlike research such as..." till the end of that passage should be omitted since this is already said in the introduction.

In the Discussion section, passage 3, the last sentence: "One possible interpretation...larger sample" should be added "larger sample of adolescents with NDDs". This sentence should be transferred to the end of passage 2 mentioned earlier in the Discussion section.

In the third passage of the Discussion, after mentioning reference 43, you should comment your results of CDI scores for NDD and non-NDD adolescents that are missing from the Results section entirely.

In the Discussion section, passage 5, the last sentence, you should add larger sample size "of adolescents with NDDs".

Kind regards.

Author Response

Response 1. In response to your comment

"In the Results section there is no CDI results presented according to NDD status". From our point of view, it is somewhat difficult to fully discern what is being requested. Our apologies but if we do not address this point could you point out specifically where you want this added. In response to your comment:

We do state (for the multivariate effect) "There was no main effect for NDDs status Wilks’ Lambda, F (4, 338.00) = 2.669, p = .032, partial η2 =.031".  We also show the univariate main effect for sex in Table 2. However, we have added a main effect for the CDI in text and adjusted the text as follows:

As can be seen in Table 2, depressive symptoms were significant (p < 0.01), with females scoring higher than males on the CDI 2: SR[S] and ...

If more is required, we can address this.

Response 2. In the Discussion section, passage 2, after mentioning reference 17, we have now omitted the sentence starting with: "Unlike research such as..." till the end of that passage since this is already said in the introduction.

Response 3. In the Discussion section, passage 3, we have now added "larger sample of adolescents with NDDs" to the last sentence ("One possible interpretation...larger sample") and transferred this to the end of passage 2 mentioned earlier in the Discussion section.

Response 4. In the third passage of the Discussion, after mentioning reference 43, we have commented on the CDI scores for NDD and non-NDD adolescents as follows:

The absence of research examining worry in adolescents with NDDs makes comparisons with the present findings difficult, however the findings regarding depressive symptoms run contrary to previous research that identified higher prevalence rates among those with NDDs [46 - 50].

Response 5. In the Discussion section, passage 5, the last sentence, we have added "of adolescents with NDDs" to larger sample size.

Reviewer 4 Report

Comments and Suggestions for Authors

Dear Authors, thank you for your clarifications and information added to the manuscript. I have one comment left for your paper and I believe it will be quite beneficial for your paper.

You explained in your reply why you repeated the psychomentric development of the PAWS and underlined why is it so different from the ways other questionnaires are developed. And I think this is very important. By I read once again section 2.2.1 and there is no this clear articulation of this importance, just pure description of the procedures and validity tests. I suggest that you highlight this difference and uniqueness of your scale in your text.

Author Response

Thank you for your comment, which we are pleased to address. In the light of your point, we have added text. This has necessitated some minor adjustment to the paragraph and also starting a new paragraph as follows:

2.2. Instrumentation

2.2.1. Adolescent Worry

The Perth Adolescent Worry Scale [PAWS, 17] is like many other instruments, having been initially generated from a review of the relevant literature and existing instruments. However, unlike other worry instruments the PAWS was co-designed via a series of focus groups with 10- to 16-year-old adolescents with or without NDDs (n=36), parents/guardians (n=16), educators (n=16), and individual interviews with school psychologists (n = 8). This rigorous procedure identified candidate items that more appropriately reflected concepts for the assessment of worry in adolescents with NDDs. Nevertheless, because of its limited use to date, it is important to further establish the psychometrics of the PAWS.

Initially, the PAWS consisted of 27 items, to…….